# Tree-rings reveal two strong solar proton events in 7176 and 5259 BCE

Nicolas Brehm [1✉], Marcus Christl [1✉], Timothy D. J. Knowles [2], Emmanuelle Casanova [2,16], Richard P. Evershed [2], Florian Adolphi [3], Raimund Muscheler [4], Hans-Arno Synal[1], Florian Mekhaldi [4,5], Chiara I. Paleari [4], Hanns-Hubert Leuschner[6], Alex Bayliss [7], Kurt Nicolussi [8], Thomas Pichler[8], Christian Schlüchter [9], Charlotte L. Pearson[10], Matthew W. Salzer [10], Patrick Fonti [11], Daniel Nievergelt[11], Rashit Hantemirov [12,13], David M. Brown[14], Ilya Usoskin [15] & Lukas Wacker [1✉]

The Sun sporadically produces eruptive events leading to intense fluxes of solar energetic particles (SEPs) that dramatically disrupt the near-Earth radiation environment. Such events have been directly studied for the last decades but little is known about the occurrence and magnitude of rare, extreme SEP events. Presently, a few events that produced measurable signals in cosmogenic radionuclides such as $^{14}C$, $^{10}Be$ and $^{36}Cl$ have been found. Analyzing annual $^{14}C$ concentrations in tree-rings from Switzerland, Germany, Ireland, Russia, and the USA we discovered two spikes in atmospheric $^{14}C$ occurring in 7176 and 5259 BCE. The ~2% increases of atmospheric $^{14}C$ recorded for both events exceed all previously known $^{14}C$ peaks but after correction for the geomagnetic field, they are comparable to the largest event of this type discovered so far at 775 CE. These strong events serve as accurate time markers for the synchronization with floating tree-ring and ice core records and provide critical information on the previous occurrence of extreme solar events which may threaten modern infrastructure.

[1] Laboratory of Ion Beam Physics, ETHZ, Otto-Stern Weg 5 HPK, 8093 Zurich, Switzerland. [2] Bristol Radiocarbon Accelerator Mass Spectrometry Facility, Bristol University, Bristol BS81TS, UK. [3] Alfred Wegener Institute, Helmholtz Centre for Polar and Marine Research, 27568 Bremerhaven, Germany. [4] Department of Geology – Quaternary Sciences, Lund University, 22362 Lund, Sweden. [5] British Antarctic Survey, Ice Dynamics and Paleoclimate, Cambridge CB3 0ET, UK. [6] Albrecht von Haller Institute for Plant Sciences, Department of Palynology and Climate Dynamics, Georg-August-University, Wilhelm-Weber-Str. 2a, 37073 Göttingen, Germany. [7] Historic England, Cannon Bridge House, 25 Dowgate Hill, London EC4R 2YA, UK. [8] Department of Geography, Universität Innsbruck, Innrain 52, 6020 Innsbruck, Austria. [9] Institute of Geological Sciences, University of Bern, Baltzerstrasse 1+3, 3012 Bern, Switzerland. [10] University of Arizona, the Laboratory for Tree-Ring Research, 1215 E. Lowell Street, Tucson, AZ 85721-0045, USA. [11] Swiss Federal Research Institute WSL, Zürcherstrasse 111, 8903 Birmensdorf, Switzerland. [12] Laboratory of Dendrochronology, Institute of Plant and Animal Ecology, Ural Branch of Russian Academy of Sciences, 8 Marta Street, 202, Ekaterinburg 620144, Russia. [13] Laboratory of Natural Science Methods in Humanities, Ural Federal University, 19 Mira Street, Ekaterinburg 620002, Russia. [14] School of Natural and Built Environment, The Queen's University, Belfast BT7 1NN, UK. [15] Sodankylä Geophysical Observatory and Space Physics and Astronomy Research Unit, University of Oulu, Oulu FIN-90014, Finland. [16] Present address: UMR7209 Archéologie et Archéobotanique: Sociétés, Pratiques et Environnements, Muséum National d'Histoire Naturelle, CP56, 43 Rue Buffon, 75005 Paris, France. ✉email: nbrehm@ethz.ch; mchristl@phys.ethz.ch; wacker@phys.ethz.ch

The Sun sporadically produces eruptive events, such as flares and coronal mass ejection, that can lead to highly intense fluxes of solar energetic particles (SEPs), which escape into the interplanetary space and possibly hit Earth[1]. When reaching the Earth, SEPs can have a dramatic impact on modern communication, navigation and power systems, satellite life expectancy, the health of astronauts, and aircraft operations[2,3]. The famous Carrington event was one of the strongest known solar storms that hit the Earth in September 1859 CE, leading to a widespread failure of the telegraph system and observable auroras all over the world[3,4]. If this, or an even larger event, occurred today the impact on global society and economy would be catastrophic[3,5].

The first instrumental observations of SEPs were conducted in the 1940s[6]. Since then their physical origin, frequency of occurrence, amplitude, and energy distribution have been studied using ground-based and space-borne data[7]. The strongest measured SEP event (called ground-level-enhancement GLE #5) took place on 23-Feb-1956 (a list of directly measured events is available at the International GLE database (https://gle.oulu.fi)). However, little is known about extreme SEP events whose very existence was unknown until a few years ago[8]. Even though observational advances such as a worldwide network of neutron monitors established in 1951/1957/1964 made it possible to characterize the Suns' eruptive behavior[9], the temporal coverage of the observational record is not long enough to assess the frequency of extremely rare but highly energetic SEP events. Statistics of sun-like stars suggest that the superflares that produce highly energetic SEPs are extremely rare[10,11].

Understanding and ultimately predicting extreme solar events will not only help in mitigating their harmful consequences on modern life and communication systems, but will also help understanding the complex magneto-hydrodynamic behavior of the Sun. In this context, cosmogenic radionuclides provide a powerful tool for reconstructing past solar activity and SEP events[12–15].

Cosmogenic radionuclides such as $^{10}Be$, $^{14}C$ and $^{36}Cl$ are mainly produced by galactic cosmic rays (GCRs) originating from outside our solar system, hitting the Earth's atmosphere. The flux of GCRs on Earth is modulated (shielded) by the geomagnetic field and the solar magnetic field carried by the solar wind[16]. As a consequence, enhanced magnetic fields (solar/heliospheric or geomagnetic) reduce cosmogenic nuclide production by GCRs on Earth and vice versa. When hitting the Earth, strong SEPs may cause an additional, short-term increase in cosmogenic nuclide production. Natural archives, such as dendrochronologically dated tree-rings ($^{14}C$) or polar ice cores ($^{10}Be$,$^{36}Cl$), are known to provide precise and temporally accurate records of past cosmogenic radionuclide production[12,17,18] giving us a unique opportunity to identify and study the characteristics of strong SEP events over the past thousands of years.

So far, three strong SEP events that led to an abrupt increase of about 1% or more in atmospheric $^{14}C$ concentrations within less than two years have been unambiguously detected over the past 3,000 years in tree-rings, and confirmed with other cosmogenic radionuclides ($^{10}Be$, $^{36}Cl$) in ice cores, in the years 993 CE, 775 CE and 660 BCE[13,14,19,20]. More, weaker, yet unconfirmed SEP events have been found in the radionuclide records[17,21,22] (Table 1). They are currently considered candidate events either because there is insufficient data coverage and precision to clearly distinguish them from normal solar modulation or because they have not yet been confirmed[23]. The above mentioned Carrington Event did not lead to a detectable increase in cosmogenic radionuclides[17,24,25] implying that associated SEPs either missed the Earth or did not have sufficient energy.

The longest absolutely dated composite tree-ring record spans the last 12,460 years[26]. At present, high-precision, annual $^{14}C$ measurements from tree-rings, which are required for the sensitive detection of strong SEP events, only cover about 2030 years[27]. Therefore, our ability to study strong SEP events is limited and theoretical estimations on their frequency[9] and energy spectra[20,28] remain highly uncertain. The extension of the time period covered with annually resolved $^{14}C$ measurements in tree-rings will provide statistically robust estimates on the magnitude and frequency of strong SEP events.

Currently, the search for strong SEP events mainly relies on existing cosmogenic nuclide records with lower (5–20 years) temporal resolution. The motivation to investigate the annually resolved $^{14}C$ signal in tree-rings between 7150 BCE and 7200 BCE came from lower resolution, synchronized[29,30] $^{10}Be$ and $^{36}Cl$ data in ice cores[31–33] with a resolution of 5 to 20 years and from decadal $^{14}C$ data (IntCal20)[27]. Another event in 5259 BCE, was discovered while investigating unexpected difficulties in producing stable Bayesian chronological models for archaeological sites and ceramic sequences dating to the period centering on the 53rd century BCE[34–36]. With hindsight, this event could not have been detected by the statistical analysis of cosmogenic nuclides from the lower resolution ice core data[37].

Here we present two previously unknown $^{14}C$ production events recorded in two (7176 BCE) and four (5259 BCE) independent, absolutely dated tree-ring chronologies. The magnitudes of the events are assessed by calculating the additional $^{14}C$ production using a global carbon cycle box model. They are compared to other known $^{14}C$ events by normalizing their magnitudes to modern geomagnetic shielding. Furthermore, the 7176 BCE event is used as a unique time marker to place a floating (i.e. not absolutely dated) portion of the Bristlecone Pine tree-ring chronology (USA) within 1–2 years on its absolutely dated part.

**Table 1 The modeled atmospheric $^{14}C$ increases, produced $^{14}C$ and normalized $^{14}C$ produced at the modern earth magnetic field is given for new and known events.**

| | Simulated $\Delta^{14}C$ increase (‰) | Additional $^{14}C$ produced (kg) | Geomagnetic field strength ($10^{22}Am^2$) Knudsen | Geomagnetic field strength ($10^{22}Am^2$) Panovska | Normalized additional $^{14}C$ produced (kg) ($M = 7.8 \cdot 10^{22}Am^2$) Knudsen | Normalized additional $^{14}C$ produced (kg) ($M = 7.8 \cdot 10^{22}Am^2$) Panovska |
|---|---|---|---|---|---|---|
| 7176 BCE (This Study) | 19.5 ± 0.6 | 28.7 ± 0.9 | 8.7 ± 1.7 | 7.5 ± 0.4 | 30.1 ± 3.6 | 28.0 ± 1.2 |
| 5410 BCE[22] | 5.6 ± 0.8 | 9.0 ± 1.1 | 7.2 ± 0.6 | 7.3 ± 0.4 | 8.6 ± 1.1 | 8.6 ± 1.1 |
| 5259 BCE (This Study) | 19.1 ± 0.6 | 29.2 ± 0.9 | 7.1 ± 0.5 | 7.4 ± 0.4 | 27.6 ± 1.4 | 28.4 ± 1.2 |
| 660 BCE[13] | 12.5 ± 1.1 | 19.2 ± 2.1 | 11.4 ± 0.6 | 9.0 ± 0.4 | 23.9 ± 2.7 | 20.9 ± 2.4 |
| 775 CE[53] | 17.6 ± 0.5 | 26.2 ± 1.0 | 10.7 ± 0.4 | 9.3 ± 0.5 | 31.5 ± 1.4 | 29.0 ± 1.4 |
| 993 CE[53] | 9.6 ± 0.6 | 14.0 ± 1.2 | 10.3 ± 0.4 | 9.0 ± 0.5 | 16.5 ± 1.5 | 15.2 ± 1.4 |
| 1052 CE[17] | 5.9 ± 1.1 | 10.1 ± 2.0 | 10.2 ± 0.4 | 9.0 ± 0.4 | 11.8 ± 2.3 | 11.0 ± 2.2 |
| 1279 CE[17] | 6.5 ± 1.6 | 9.2 ± 2.7 | 9.6 ± 0.3 | 9.2 ± 0.5 | 10.4 ± 3.0 | 10.1 ± 2.9 |

## Results

Atmospheric $^{14}C$ concentrations measured in several independently built tree-ring chronologies increased by nearly 2% within two years in 7176 BCE and 5259 BCE (Fig. 1, Table 1, Supplementary Fig. 1) thereby exceeding the rise in $^{14}C$ concentration of the largest known event in 775 CE[14]. For the 7176 BCE event (Fig. 1a) an average increase of $(19.5 \pm 0.6)$‰ in $^{14}C$ concentrations was observed in the two existing absolutely dated tree-ring chronologies from central Europe covering that time range, the German oak and the Eastern Alpine conifer[38,39] chronologies. The 7176 BCE event was also found in a currently floating portion of the Bristlecone Pine chronology from the USA (Supplementary information S1). The 5259 BCE event (Fig. 1b) was found in four absolutely dated tree-ring chronologies from the Alps, Ireland, Russia and the USA showing an average increase in atmospheric $^{14}C$ concentrations of $(19.1 \pm 0.6)$ ‰. A potentially early increase is observed in the Bristlecone Pine in 5260 BCE.

Total and excess $^{14}C$ productions from the SEP event were calculated using a global carbon cycle model[17] (Methods). The excess production (and related uncertainty) was calculated using a Monte–Carlo approach by fitting offset and amplitude of a Gaussian-shaped production spike to the data (Fig. 2, Methods). For each of the 1000 different Monte–Carlo data realizations the amount of additional $^{14}C$ produced was extracted. The average amount of $^{14}C$ excess produced during the two events was $(29.2 \pm 0.9)$ kg for the 5259 BCE event and $(28.7 \pm 0.9)$ kg for the 7176 BCE event (Table 1). This is more than (7176 BCE) or comparable to (5259 BCE) the amount of $^{14}C$ produced by the strongest $^{14}C$ event detected so far (775 CE, $(26.2 \pm 1.0)$ kg excess $^{14}C$). In both cases, a modeled duration of 0.3 year, 2-sigma of the Gaussian, was used to fit the measured $^{14}C$ increase. This is significantly less than the tree-ring resolution of one year.

Excess $^{14}C$ production of all known events (Supplementary Fig. 3) was (re-)calculated using the above described procedure

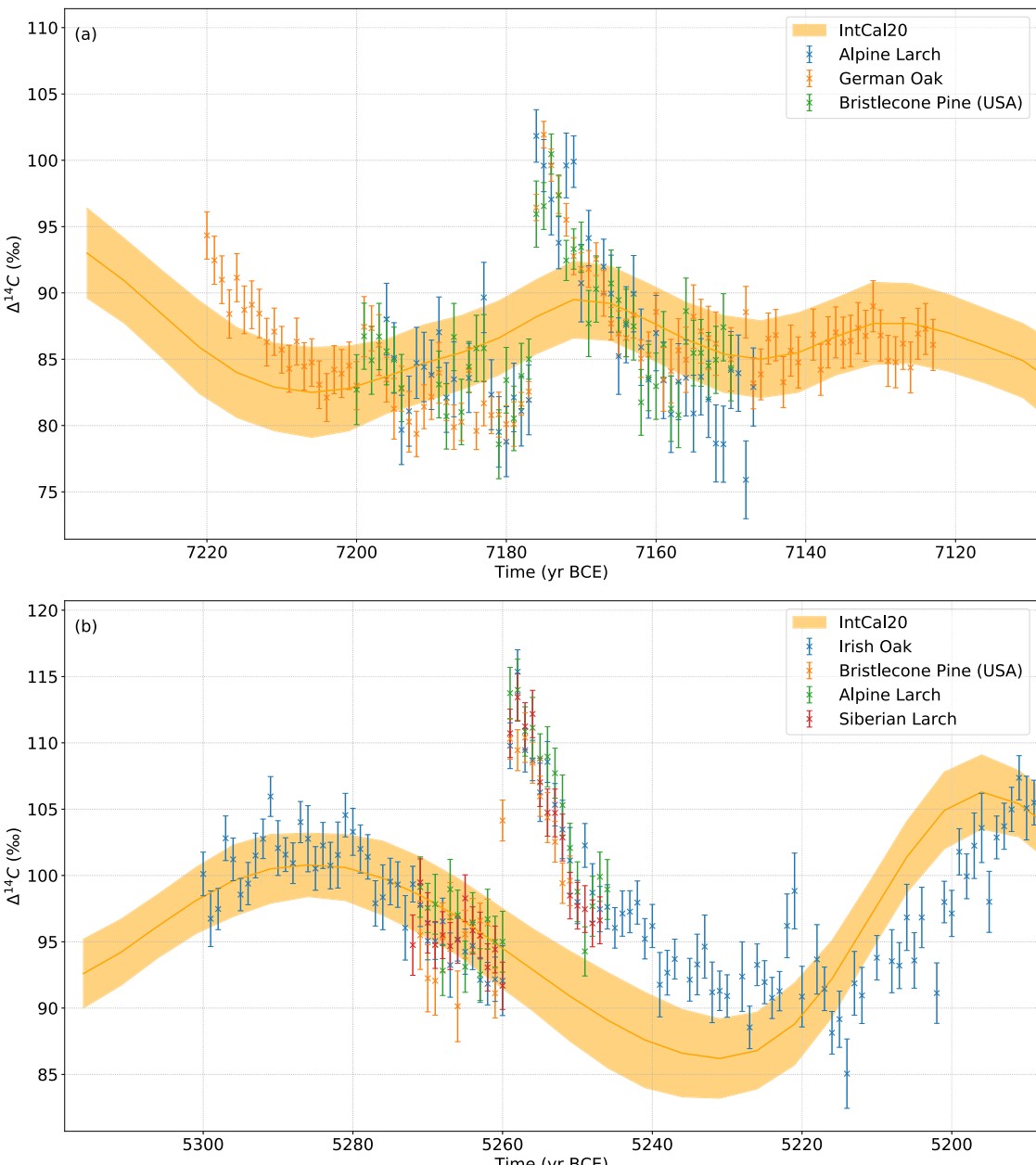

**Fig. 1 $^{14}C$ data of the two events compared to IntCal20.** Annual $^{14}C$ concentrations reported as $\Delta^{14}C$ with 1-σ errors from different trees for the two found events (7176 BCE (**a**), 5259 BCE (**b**)) in comparison with the IntCal20 calibration curve[27] (orange band).

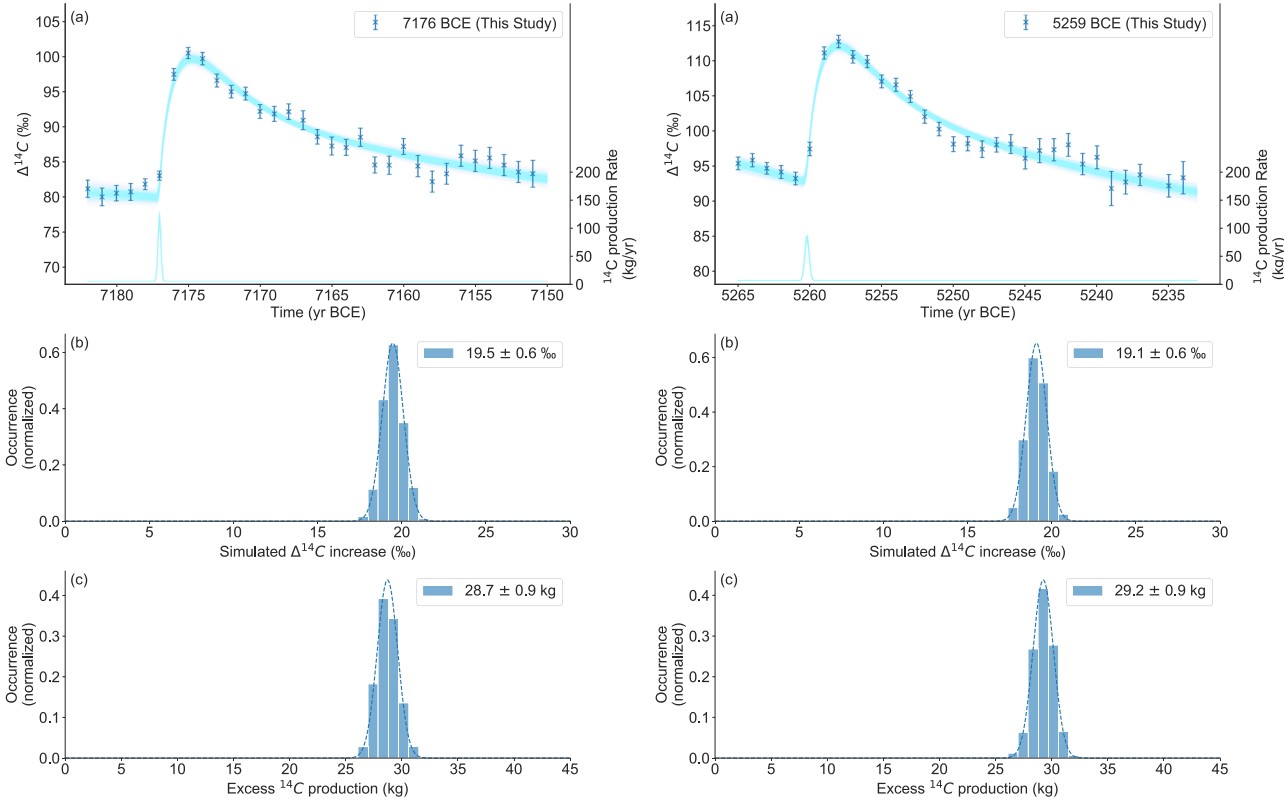

**Fig. 2 Evaluation procedure of the found events. a** Mean data of the two events (7176 BCE left, 5259 BCE right) with 1-σ errors and result of 1000 Simulations (blue lines) The fitted Gaussian shaped production spikes for all simulations is also shown. **b** Distribution of the simulated $\Delta^{14}C$ increases (blue bars) with a gaussian fit (dashed line). **c** Distribution of excess $^{14}C$ production (blue bars) with Gaussian fit (dashed line).

(Fig. 2a). When comparing different events, it is, however, important to note that the same event (in terms of the proton flux and energy distribution) can lead to different excess productions of cosmogenic radionuclides depending on the strength of the Earth's geomagnetic field at the time of the event. A weak geomagnetic field would result in higher excess production of cosmogenic radionuclides by the SEP event and vice versa. We therefore recalculated the excess of $^{14}C$ production by normalizing all events relative to the modern geomagnetic dipole moment of $7.8 \cdot 10^{22}$ Am$^2$ using two different geomagnetic field records[40,41] (see Methods/ Supplementary Figs. 4 and 5 (Fig. 3). The resulting normalized excess $^{14}C$ productions for the 7176 BCE event are not significantly different from the original value (Table 1). The 5259 BCE event occurred during the period of weaker magnetic field strength and, thus, referring to modern geomagnetic shielding the excess $^{14}C$ production would be slightly lower. The normalization procedure also significantly changes the excess $^{14}C$ production of the other known events. For the 775 CE event, the two geomagnetic field reconstructions give different results, while the Knudsen[41] reconstruction leads to a 20% increase of excess $^{14}C$, using the Panovska[40] reconstruction increases it by only 10%. When comparing the events after normalization we observe that the 7176 BCE and the 5259 BCE events are of similar, but slightly weaker magnitude than the 775 CE event. In any case, they are significantly stronger than both 660 BCE and 993 CE events.

## Discussion
The two identified events are assigned to the years 5259 BCE and 7176 BCE in several independent and absolutely dated tree-ring chronologies. The fact that the events are accurately dated to one

year makes them (and all other $^{14}C$ events) excellent time markers for the synchronization of chronologies. For example, an older floating portion of the USA Bristlecone chronology could not be comprehensively connected to its master chronology using tree-ring dating techniques, due to a very short (not statistically verifiable) period of overlap between the records. Instead, this chronology was assigned possible calendar dates based upon a tentative dendrochronological match within a broader time window suggested by conventional radiocarbon dating[42] (see SI for full details). The results of this analysis clearly showed the 7176 BCE event, placing this previously floating sequence on an absolute timescale. Further substantiation of this connection will provide an extension of the Methuselah Walk bristlecone pine chronology beyond 6827 BCE (where the calendar dated portion ends) across a full 10,399 year sequence.

For the 5259 BCE event, we note that the bristlecone record shows a potential early increase in 5260 BCE. There are a number of possible causes for this. Very narrow ring widths (<0.5 mm) during this period, indicating difficult environmental conditions, may have hampered the complete dissection of the rings, or there may be a previously undetected dating error in this earlier calendar dated portion of the record, where cross-checking with other site chronologies is not possible. Alternatively, this anomalous result may relate to regional shifts in growing season or physiological differences between deciduous and indeciduate trees. If, for example, the $^{14}C$ event occurred in late summer or autumn of 5260 BCE, i.e. after the end of the ring formation period in Ireland, the Alps and the Russian north, but before the end of tree ring formation in California, this would result in increased $^{14}C$ content of the 5260 BCE bristlecone pine tree ring, but not in the other chronologies. However, this cannot be argued on the basis of published information on the end of today's

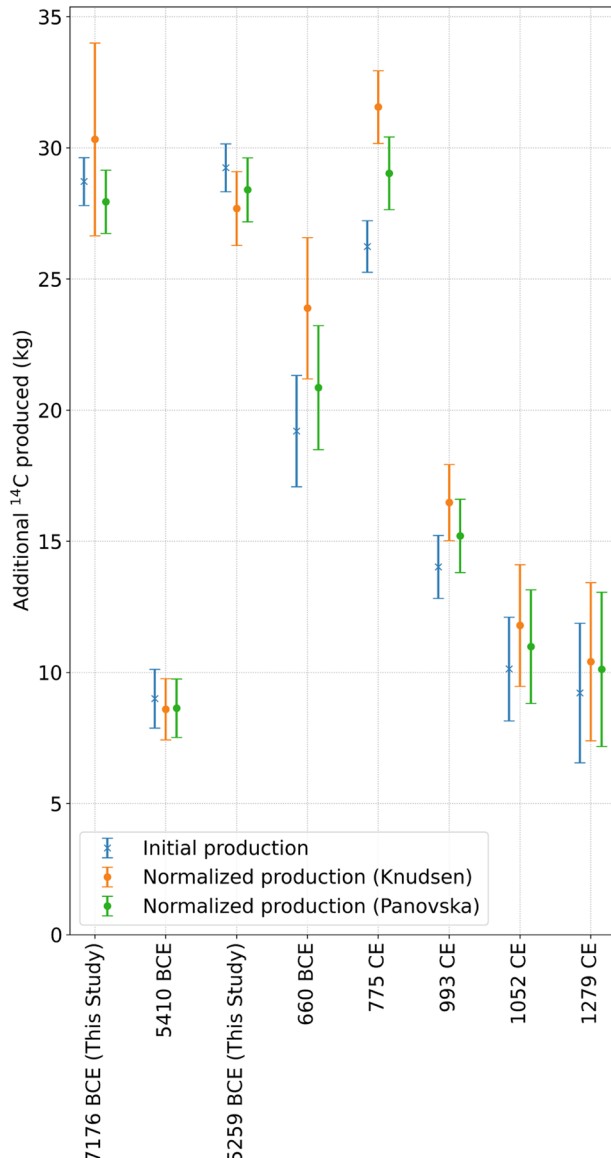

**Fig. 3 Initial and normalized event productions.** Excess $^{14}C$ produced during the events (blue) in comparison to the excess $^{14}C$ production normalized to today's geomagnetic field strength using two different records (orange Knudsen[41], green Panovska[40]) with 1-σ error bars for all known $^{14}C$ events.

Polar ice cores, which are regarded as extremely valuable climate archives, have typically been dated with a precision of about 10 years within the Holocene[29] and even less in the transition to the glacial period[42]. The absolutely dated $^{14}C$ events in tree-rings can now be used to synchronize the ice core chronologies via cosmogenic $^{10}Be$ and $^{36}Cl$ resulting in a precision of about 1–2 years limited by several factors such as weather noise, sampling, and radionuclide transfer times to the ice core[43]. As a result, chronologies that so far were synchronized with absolutely dated tree-ring records using the low resolution structure of existing $^{14}C$ or $^{10}Be$ records[29] can now be placed within 1–2 years or about 10 times more precisely around these events using the abrupt change in radionuclide concentrations caused by SEP events.

The increasing number of discoveries of strong SEP events that hit Earth over the past 12,000 years indicates that they cannot be considered as extremely rare anymore. So far, only a few sequences covering a total of 2030 yrs of the past 12,400 years (16.5%) have been analyzed with $^{14}C$ at the annual or biennial resolution, which is required to unambiguously detect SEP events, revealing five strong SEP events (Fig. 4). These findings might lead to the conclusion that strong SEP events hit Earth once every about 400 years on average (i.e. five events in 2030 years). This simple statistic, however, is likely biased by the fact that some studied time periods were not randomly selected for annual analysis, but rather were targeted based on indications from lower-resolution cosmogenic datasets or archaeological evidence that the present multi-annual $^{14}C$ record might contain more structures than previously visible. In an unlikely case that all major events have already been found over the last 12,400 years, the lower limit of occurrence for strong SEP events can be estimated as five events in 12,400 years or one every ~2400 years. Based on our findings we can constrain the occurrence rate of strong solar events to one every 400–2400 years. We nevertheless expect more events to be discovered as additional annually resolved data becomes available, leading to more precise estimates of the frequency of occurrence and magnitude of strong SEP events. The estimate of the occurrence rate of strong SEP events shows that they are more frequent than suggested by the revised statistics of superflares on sun-like stars (once every 3000--6000 yr)[11]. However, a direct relation between superflares and strong SEP events remains unknown. The statistical constraints provided by the data will help to test different approaches in solar/stellar physics describing the occurrence and magnitude of extreme events in sun-like stars. Ranking among the three largest short-term $^{14}C$ production events, the impact of the newly discovered events would have been catastrophic for aircraft, satellites, modern telecommunication and computer systems[44–46], if they occurred today.

## Materials and Methods

**Sample preparation and measurement.** For the 7176 BCE event dendrochronologically dated wood samples from Ireland, supplied by the University of Belfast, and from the Alps, supplied by the University of Innsbruck, were dissected into annually resolved samples weighing 30–60 mg.

For the analyses performed at ETHZ, typically 54 tree-ring samples with four wood blanks (2 BC and 2 KB) and 2 1515 CE reference samples[47] each weighing 30–60 mg, were prepared in 15 ml glass test tubes together in a batch (making 60 in total). In a slightly modified procedure following Němec et al.[48], samples were first soaked in 5 ml 1 M NaOH overnight at 70 °C in an oven. Then the samples were treated with 1 M HCl and 1 M NaOH for 1 hour each at 70°C in a heat block, before they were bleached at a pH of 2–3 with 0.35 M NaClO$_4$ at 70 °C for 2 h. The remaining white holo-cellulose was then freeze-dried overnight.

About 2.5 mg dried holo-cellulose was wrapped in cleaned Al capsules[49] and converted to graphite using the automated graphitization line AGE[50]. A measurement set was made up of three oxalic acid one (OX1) and four oxalic acid two (OX2) standards, 27 samples, two cellulose blanks, two chemical blanks, and two 1515 CE reference samples (individual cellulose preparations of the 1515 CE reference is used for at least two measurements) and measured in the MICADAS

growing seasons (Bristlecone pine—late August[43,44], Alpine larch —October[45], Irish oak – October[46], Siberian larch-late August). Furthermore, the fact that the event occurred during the Holocene climatic optimum, which was characterized by a weaker latitudinal temperature gradient[47], may mean that more synchronous growing seasons would be more likely. Another possibility may relate to the fact that Bristlecone Pine is the only indeciduate tree in this record. Deciduous trees such as oak and larch store photosynthates produced during the end of growing season to grow the next year's earlywood[48]. If the $^{14}C$ event occurred towards the very end of the deciduous tree's growth season it is possible that only the indeciduate Bristlecone Pine would register the change in the same year. If this early increase of $^{14}C$ in Bristlecone Pine can be confirmed by future replicate measurements, the timing of the event would be confined precisely to the end of summer/autumn 5260 BCE."

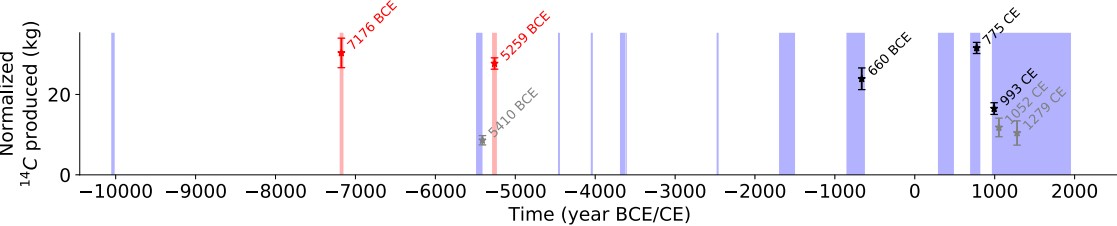

**Fig. 4 Magnitude and occurrence of all known [14]C events over time.** Shaded regions mark time periods where IntCal20[27] (purple) and our data (red) are based on annual or biennial resolution. Data points show the timing of all known (black) and newly identified (red) [14]C events and their normalized additional production with 1-σ errors. Event candidates are indicated in grey.

### Table 2 Statistical analysis of the repeated measurements of the different chronologies.

|  | $\chi^2$ | Deg. of freedom | $P(\chi^2>)$ value |
|---|---|---|---|
| All (7176 BCE) | 255.3 | 202 | 0.01 |
| Alpine Larch (7176 BCE) | 6.3 | 6 | 0.39 |
| German Oak (7176 BCE) | 112.6 | 93 | 0.08 |
| Bristlecone Pine (7176 BCE) | 12.4 | 13 | 0.50 |
| All (5259 BCE) | 254.0 | 216 | 0.04 |
| Irish Oak (5259 BCE) | 89.0 | 82 | 0.28 |
| Alpine Larch(5259 BCE) | 15.9 | 13 | 0.25 |
| Siberian Larch (5259 BCE) | 10.2 | 24 | 0.99 |
| Bristlecone Pine (5259 BCE) | 16.6 | 24 | 0.87 |

accelerator mass spectrometer[51]. Two measurement sets were typically prepared from one set of samples within a week and subsequently measured. A second graphite sample was subsequently prepared and measured from one third of the prepared cellulose samples for quality control purposes.

Two internal wood reference materials from 1515 CE (Pine and Oak) and two different radiocarbon free wood blanks (*Kauri Stage 7, KB*, and *Brown Coal, BC*, from *Reichwalde*) were repetitively analyzed together with the annual samples. While the wood-blanks were used for blank subtraction in the data evaluation process, the 1515 CE references were used for quality control only.

For the analyses performed in Bristol, tree ring samples (20–30 mg) were processed alongside two KB wood blanks and two oak 1524 CE reference samples in 15 ml glass culture tubes. Samples were pretreated following the standard BRAMS wood pretreatment method (code BABAB; following Němec et al.), as described by Knowles et al.[52]. Around 2.5 mg dried holo-cellulose was combusted, graphitized, and measured using the Bris-MICADAS AMS. Full details of the procedures employed are described by Knowles et al.[52].

Several samples were repetitively measured, and a chi-squared analysis showed that the repeated measurements were generally in good agreement with one another. The resulting chi-squared values of the different chronologies are shown in Table 2. Small observed differences are likely due to wood anatomical effects or due to the limitations of cutting tree-rings accurately for species with narrow tree-rings (Supplements S1). The high overall $\chi^2$ value events are likely due to regional offset[53] and the earlier increase signal in the Bristlecone Pine chronology in the 5259 BCE event.

**Modeling**. To model the carbon cycle an improved carbon box model based on the model of Güttler et al.[54] was used (Supplementary Fig 2). The model of Güttler uses 11 Boxes to simulate the exchange between the global atmosphere biosphere and Oceans. To model the northern and southern hemispheres separately our model was extended to 22 boxes (11 boxes for each hemisphere). The carbon content of each box is distributed according to the respective relative carbon reservoir masses of the corresponding hemisphere. Radiocarbon is produced in the stratosphere and the troposphere of both hemispheres, where 70% is produced in the Stratosphere and 30 % in the Troposphere. The fluxes were adjusted to ensure a correct $\Delta^{14}C$ offset between the northern and southern troposphere. Seasonal variability of fluxes was not considered in the model.

The [12]C and [14]C content of each box after a time step is calculated with

$$N_i^{12}(t + \Delta t) = N_i^{12}(t) + dN_i^{12}(t)\Delta t \tag{1}$$

$$N_i^{14}(t + \Delta t) = N_i^{14}(t) + dN_i^{14}(t)\Delta t \tag{2}$$

$$dN_i^{12}(t) = \sum_j F_{ji}^{12}(t) - \sum_j F_{ij}^{12}(t) \tag{3}$$

$$dN_i^{14}(t) = -\lambda N_i + \sum_j F_{ji}^{14}(t) - \sum_j F_{ij}^{14}(t) + P_i(t) + P_{st,i} \tag{4}$$

Here $N_i^{12,14}$ is the $C^{12,14}$ content of each box in Gt and $\lambda$ is the decay constant of [14]C. The time step $\Delta t$ was chosen to be one month for all the following simulations. The [14]C and [12]C fluxes are given by the following:

$$F_{ij}^{12}(t) = F_{ij,st}\frac{N_i^{12}(t)}{N_{i,st}^{12}}, F_{ij}^{14}(t) = F_{ij,st}\frac{N_i^{12}(t)}{N_{i,st}^{12}}\frac{N_i^{14}(t)}{N_i^{12}(t)}\frac{m_{14}}{m_{12}} = F_{ij,st}\frac{N_i^{14}(t)}{N_{i,st}^{12}}\frac{m_{14}}{m_{12}} \tag{5}$$

Here $F_{ij,st}$ are the fluxes given in Supplementary Fig. 2. The fluxes are scaled depending on the deviation from the Holocene steady state $N_{i,st}^{12}$. The steady state was computed by simulating 200'000 years with the constant production rate $p_{st} = 1.76\frac{at}{cm^2 s}$. The model does not consider any isotopic fractionation and thus the [14]C fluxes scale just as the [12]C fluxes.

With this a general expression for all boxes at any time can be achieved.

$$N(t + \Delta t) = N(t) + (\Lambda N(t) + F^T(t)1 - F(t)1 + P(t) + P_{st})\Delta t \tag{6}$$

$$F = \begin{bmatrix} F^{12}(t) & 0 \\ 0 & F^{14}(t) \end{bmatrix}, \Lambda = \begin{bmatrix} 0 & 0 \\ 0 & -\lambda \end{bmatrix}, N(t) = \begin{pmatrix} N^{12} \\ N^{14} \end{pmatrix}, 1_i = 1 \text{ forall } i \tag{7}$$

$$P(t) = Vp(t), P_{st} = Vp_{st}$$

$$V_i = \begin{cases} 0.5 \cdot 0.7, & \text{if } i = \text{Stratosphere } N, S \\ 0.5 \cdot 0.3, & \text{if } i = \text{Troposphere } N, S \\ 0, & \text{else} \end{cases} \tag{8}$$

The $\Delta^{14}C$ of each box for the simulation is calculated by the following expression:

$$\Delta^{14}C_i(t) = \frac{\frac{N_i^{14}(t)}{N_i^{12}(t)} - \frac{N_{TN,st}^{14}}{N_{TN,st}^{12}}}{\frac{N_{TN,st}^{14}}{N_{TN,st}^{12}}} \cdot 1000 \tag{9}$$

Where $N_{TN,st}^{12,14}$ are the steady state [12,14]C contents of the northern troposphere.

To get a reasonable start state at a given time the production rate for the whole IntCal record has been calculated and the simulated state can be loaded at any time before 1950 CE.

The events were evaluated by generating 1000 realizations of the data distributed with the errors of the measurements. For each realization a Gaussian shaped production spike was fitted and the additional production was extracted by integrating the Gaussian production. Since the energy spectrum of the SEPs is softer than the one of the galactic cosmic rays the excess production of the event was mainly produced in the stratosphere (90%) and only 10% in the troposphere. The errors were estimated by using the widths of the resulting distributions.

**Geomagnetic field correction**. Since the production of [14]C in the Earth's atmosphere is affected by geomagnetic shielding of the flux of SEPs, this effect needs to be accounted for in such a way that all the events are normalized to the same reference standard, specifically to the modern conditions with the dipole moment $M = 7.8 \cdot 10^{22}$ A·m[2] [55]. The geomagnetic field has a very complex structure at the Earth's surface[56], but for energetic particles, the dipole moment is the most important since higher momenta decay faster with distance, and the eccentric +tilted dipole approximation is widely used[57]. Since the true dipole moment is difficult to determine in the past, before the era of direct measurements, an approximation of the virtual axial dipole moment (VADM) is reconstructed from paleo- and archeo-magnetic models[58]. Accordingly, the geomagnetic-shielding effect on energetic particles is often quantified via the VADM value of $M$. The modern (epoch 2000) dipole moment is $M_0 = 7.8 \cdot 10^{22}$ A·m[2]. Here we normalize the apparent [14]C production $Q$ during different events to the standard modern conditions, viz. the production that would be if the event took place nowadays. For

that, we modeled the dependence of $Q$ on $M$ using a broad range of SEP spectra measured during the recent decades[59]. The spectra are bound between the hardest known spectrum of SEP event on 23-Feb-1956 (GLE 05[60]) and the softest strong-event spectrum of 04-Aug-1972 (GLE 24[60]), as shown in Supplementary Fig. 4a. The same plot shows also the global yield function, denoted as $Y_p(E)$, of $^{14}$C for three VADM values of $M = 3, 6$ and $9$ (x10$^{22}$ A·m$^2$). The global production function is defined as

$$Y_p(E, M) = \int_0^{\pi/2} Y(E) \cdot H(E, M, \theta) \cdot \sin(\theta) \cdot d\theta \quad (10)$$

where $Y(E)$ is the columnar yield function of $^{14}$C by protons with energy $E$ in the Earth's atmosphere[61], $H(E)$ is a step-like magnetospheric transmissivity function taking the value of 0 for $E < E_c$ and 1 otherwise, where $E_c$ is the energy corresponding to the local geomagnetic rigidity cutoff defined by the dipole moment $M$ and the polar angle (co-latitude) $\theta$. The production is higher for a weaker geomagnetic field (smaller values of $M$) and vice versa. The production of $^{14}$C by a specific SEP event with a given proton spectrum $J_p$ is defined as the integral of the product of the yield function and the energy spectrum of particles.

$$Q(M) = \int_0^\infty Y_p(E, M) \cdot J_p(E) \cdot dE \quad (11)$$

The dependence of $Q$ on $M$ is shown in Supplementary Fig. 4b for the two analyzed SEP spectra. In order to compensate for the variable geomagnetic field, we introduce the correction factor, which relates the $^{14}$C production at the VADM $M$ to that at the reference field $M_0$:

$$C = \frac{Q(M_0)}{Q(M)} \quad (12)$$

The correction factor is shown in Supplementary Fig. 4c for the two SEP spectra. Although the spectra are essentially different (panel a) and the isotope production for the two events is different too (panel b), the correction factor appears robust against the exact spectral shape. This correction factor was further applied to compare the strength of all studied events to the standard conditions, as shown in Table 1.

Reconstructions of the geomagnetic field in the past is a difficult task, and the result can be uncertain. To cover a possible range of VADM values we took a conservative approach and considered two paleomagnetic reconstructions for the last millennia, as shown in Supplementary Fig. 5: a recent GGF100k paleomagnetic model by Panovska et al.[40] and a reconstruction by Knudsen et al.[41] based on the GEOMAGIA50 database. Although there are many other paleomagnetic reconstructions, these two cover the full range of uncertainties.

The uncertainties of the corrected for the VADM changes $^{14}$C production (last two columns in Table 1) are mostly defined by the uncertainties of the actual production (column 3) and VADM uncertainties (columns 4 and 5). The resultant uncertainty of the corrected production was assessed using the Monte–Carlo method by randomly picking pairs of $M$ and $Q$ from the normally distributed values within the ranges shown in Table 1 and computing the mean and the standard deviation of the corrected $Q$ values.

## Data availability

The excel data that support the findings of this study are available in PANGEA at doi. Source data are provided with this paper.

## Code availability

Any computer code used for evaluation of the results of this study will be available on request.

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

## Acknowledgements

The Laboratory of Ion Beam Physics is partially funded by its consortium partners PSI, EAWAG, and EMPA. N.B. is funded by the Swiss National Science Foundation (SNSF grant #SNF 197137). The establishment of the BRAMS Facility was jointly funded by the NERC, BBSRC and the University of Bristol and the measurements in this work were partly funded by an ERC Proof of Concept grant awarded to R.P.E. and financing E.C. postdoctoral contract (LipDat H2020 ERC-2018-PoC/812917). We thank Bisserka Gaydarska for sub-sampling the inter-laboratory replicates from M49, M234, Q2729 and Q2750, Cathy Tyers for reviewing the dating of the Irish and German samples, and Alexander Land for assistance in dating sample M49. P.F. received funding from the SNF Sinergia project CALDERA (no. 183571). R.H. is funded by Russian Science Foundation (grant # 21-14-00330). I.U. acknowledges the support from the Academy of Finland (grant 321882 ESPERA). C.L.P.'s and M.W.S.'s work on bristlecone pine was funded by the M.H. Wiener Foundation (ICCP Project). K.N. acknowledges the support provided by the Austrian Science Fund FWF (grant I-1183-N19).

## Author contributions

L.W., M.C. and N.B. designed the study with input from H.-A.S., R.M. and F.A., A.B., D.B., H.-H.L., K.N, T.P., C.S., C.L.P., M.W.S., D.N., E.C., P.F., R.P.E. and R.H. supplied the annually resolved tree-ring samples and are responsible for the documentation of the dendrochronology. Radiocarbon measurements and analyses were performed by N.B., L.W. and T.K. The modeling and interpretation of the 14C data were done by N.B, M.C., and L.W. with input from I.U., F.A., R.M., H.-A.S., F.M. and C.L.P., N.B, M.C., and L.W. wrote the manuscript with input from all other authors.

## Competing interests

The authors declare no competing interests.
