## [Peer Review File · Nature Communications]

Tree rings reveal two strong solar proton events in 7176 and 5259 BCEReviewers' Comments:

Reviewer #1:

Remarks to the Author:

This paper presents two newly found ¹⁴C excursions in 7176 and 5259 BCE, suggesting possible occurrence of major solar energetic particle (SEP) events in the mid-Holocene. The authors rigorously estimate the intensity of the events, as well as the events previously found, by applying the reconstruction of geomagnetic field intensity. Occurrence rate of large SEP events as well as the information on the maximal intensity of SEP contribute to the understanding of solar physics and the space weather research. Two large SEPs found in this study increase the statistics and make a great advance. The absolutely dated events also contribute as the time markers of every kind of sedimentological works, including the ones using tree rings and ice cores. Generally, the manuscript is well written, although some improvements are needed to make the arguments more explicit.

Followings are the detailed comments.

Major comments:

1. The writing/discussions in Line 176-192 need some improvement. First, it is not clear how "once every about 400 years on average" was derived from the five events in 8000 years shown in Figure 4. Second, the authors state "This greatly improves the existing estimates based on only a few extreme events known until recently" and I assume that this sentence points to the improved statistics and the estimates on the maximal SEP intensity, but the sentence "Ranking..." in between makes the arguments unclear. Third, there need some additional statement/specific number what are the statistics suggested by the observation of sun-like stars and how much it disagrees with the statistics improved by this study. When making this statement, make it clear how the occurrence rate of large SEPs was derived in this study as pointed above.

2. There is some discrepancy between the records obtained using the Bristlecone Pine and the Oak/Larch regarding the 5259 BCE event. The authors suggest that this discrepancy may have been caused by the difficulty in separating the adjacent rings of pine tree. However, there could be another possible explanation for this discrepancy. I suggest to include the discussion on the point if the difference may or may not be caused by the difference in the species of the trees used. Oak and Larch are deciduous tree, while pine tree is indeciduate. It is therefore possible that they grow rings using photosynthetic product from different season/period. If the authors find that the discrepancy between the records may be explained by the difference in the species, make further discussion on what is the possible timing (year and season) of this SEP event. The 22-box carbon cycle model used in this study allows the monthly-resolved calculation, and it may help giving a strong constraint on the timing.

Minor comments:

Line 76-78: Here the authors cite the paper by Miyake et al. (2017) and regard the 5480 BC event as "currently considered candidate events", but not listed in Table 1 or discussed in the manuscript. I suggest to briefly note the condition of this event here and why it was excluded from the analyses. Also, I suggest to cite Table 1 here to make it clear which events the authors regard as "candidates".

Line 82-84: The continuous record may only be available for the past 2030 years; however, there are a few floating records as are shaded in Figure 4 or some other records that are sensitive enough to detect past SEPs but not included in IntCal20. Please give a brief but comprehensive overview about the availability of the high-precision data.

Line 89-90: It is not clear, especially to the readers who are not familiar with this field, if the 5-10 yr resolution record is the same as the above mentioned 12,460-yr long record. Include the information on the span of the record if the authors meant any specific record.

Line 159: It was not clear based on what the specific number "beyond 6827 BCE" came from.

Other comments:

1. The manuscript is overall well written; however, I'm concerned about the overuse of parentheses. There are even places where parentheses are unnecessary. Avoid using parentheses and write down in concrete terms. There are places it is not easy to understand.
2. Figure 4: Indicate the year of the events, otherwise it is difficult to identify which of the events the data points are denoting. Please also increase the number of ticks on the X-axis.
3. Line 122 and some others: (amount of) excess 14C -> (amount of) 14C excess
4. Line 134: excess 14C production -> excess of 14C production
5. Line 277, 287, 291, and 298: EDFig.4a -> Supplementary Fig.4a?

Reviewer #2:

Remarks to the Author:

This paper reports observations of 14C increases indicating major solar energetic particle events at two historical periods. The paper is well written and comprehensive. The methods and analysis are appropriate, well executed, and clearly described. It is an important contribution to the field. I find no significant problems with the manuscript and I believe it should be published with only a few minor edits, noted below.

On lines 277, 287, and 291 there are figures referred to by "EDFig"; these refer to figures in the supplementary material and I do not know what is meant by "ED".

Reference 8 appears to be missing information and should be completed.

Detailed reply to the reviewer's comments

The reviewer's comments are displayed in black while our replies are written in blue.

Reviewer #1 (Remarks to the Author):

This paper presents two newly found ^{14}C excursions in 7176 and 5259 BCE, suggesting possible occurrence of major solar energetic particle (SEP) events in the mid-Holocene. The authors rigorously estimate the intensity of the events, as well as the events previously found, by applying the reconstruction of geomagnetic field intensity. Occurrence rate of large SEP events as well as the information on the maximal intensity of SEP contribute to the understanding of solar physics and the space weather research. Two large SEPs found in this study increase the statistics and make a great advance. The absolutely dated events also contribute as the time markers of every kind of sedimentological works, including the ones using tree rings and ice cores. Generally, the manuscript is well written, although some improvements are needed to make the arguments more explicit. Followings are the detailed comments.

We thank the Reviewer very much for this supporting review and the very useful comments. We believe that we have carefully addressed all the comments. (see below).

Major comments:

1. The writing/discussions in Line 176-192 need some improvement. First, it is not clear how "once every about 400 years on average" was derived from the five events in 8000 years shown in Figure 4. Second, the authors state "This greatly improves the existing estimates based on only a few extreme events known until recently" and I assume that this sentence points to the improved statistics and the estimates on the maximal SEP intensity, but the sentence "Ranking..." in between makes the arguments unclear.

We thank the Reviewer for pointing out this unintended ambiguity.

The simple estimation of an average event rate of one per 400 years was derived by dividing the about 2000 years of annually resolved measurements in IntCal20 by the 5 events found so far. We make this clearer now. In the revised version we additionally estimate the lower boundary for the frequency of occurrence of one event every 2400 years. (5 events / 12,400 years) We now write:

"The increasing number of discoveries of strong SEP events that hit Earth over the past 12,000 years indicates that they cannot be considered as extremely rare anymore. So far, only a few sequences covering a total of 2030 yrs of the past 12,400 years (16.5%) have been analyzed with ^{14}C at the annual or biennial resolution, which is required to unambiguously detect SEP events, revealing five strong SEP events (Figure 4). These findings might lead to the conclusion that strong SEP events hit Earth once every about 400 years on average (i.e. five events in 2030 years). This simple statistic, however, is likely biased by the fact that some studied time periods were not randomly selected for annual analysis, but rather were targeted based on indications from lower-resolution cosmogenic datasets or archaeological evidence that the present multi-annual ^{14}C record might contain more structures than previously visible. In an unlikely case that all major events have already been found over the last 12,400 years, the lower limit of occurrence for strong SEP events can be estimated as five events in 12,400 years or one every ~2400 years. Based on our findings we can constrain the

occurrence rate of strong solar events to one every 400 - 2400 years. We nevertheless expect more events to be discovered as additional annually resolved data becomes available, leading to more precise estimates of the frequency of occurrence and magnitude of strong SEP events.”

Third, there need some additional statement/specific number what are the statistics suggested by the observation of sun-like stars and how much it disagrees with the statistics improved by this study. When making this statement, make it clear how the occurrence rate of large SEPs was derived in this study as pointed above.

We thank the Reviewer for this important comment.

In the revised version we refer to the significantly revised occurrence rate of superflares from sun like stars based on ref [11]. According to their tightened definition of sun-like stars, the occurrence rate of superflares is estimated as one per star per 3000--6000 years. It is, however, presently unknown, due to a lack of relevant theoretical models, how exactly superflares can be related to extreme SEPs. This will require the development of new self-consistent models of particle acceleration and transport in extreme conditions at and near the Sun, that lies beyond the scope of the present experimental work.

As suggested by the reviewer, we added a short discussion on this topic comparing our results with observations of sun-like stars. The following sentences were added:

“The new estimate of the occurrence rate of strong SEP events shows that they are more frequent than suggested by the revised statistics of superflares on sun-like stars (once every 3000--6000 yr)[11]. However, a direct relation between superflares and strong SEP events remains unknown.”

2. There is some discrepancy between the records obtained using the Bristlecone Pine and the Oak/Larch regarding the 5259 BCE event. The authors suggest that this discrepancy may have been caused by the difficulty in separating the adjacent rings of pine tree. However, there could be another possible explanation for this discrepancy. I suggest to include the discussion on the point if the difference may or may not be caused by the difference in the species of the trees used. Oak and Larch are deciduous tree, while pine tree is indecudate. It is therefore possible that they grow rings using photosynthetic product from different season/period. If the authors find that the discrepancy between the records may be explained by the difference in the species, make further discussion on what is the possible timing (year and season) of this SEP event. The 22-box carbon cycle model used in this study allows the monthly-resolved calculation, and it may help giving a strong constraint on the timing.

This is an important comment and indeed it would be very interesting to explore a potential difference between species further. We however think that this is not the reason for the observed discrepancy. In Büntgen et al. 2018, data from bristlecone pine from the White Mountains of California show the same response to the 775 CE and the 993 CE events as oaks, larch and other species. So, for these two time periods, any such possibilities can be ruled out. For this study we are working in a more ancient time frame and the only bristlecone samples available come from a high altitude location where growth is represented by very narrow ring widths, which makes the dissection very challenging.

Given that the results on two separate dissections of material from the 5259 BCE event produced slightly differing results, it currently seems most probable that the problem here is incomplete dissection due to highly resinous and narrow rings. Regarding the clearness of response for the 7176 BCE event in bristlecone from the same location, it again seems that the problem is more likely associated with the practicalities of dissection or dating than physiology, growth season and timing of the actual SEP.

We now write:

“We do not consider this likely to result from growth season or physiological differences between the trees used in this study because the 775 CE and 993 CE events were consistently recorded in bristlecone pine compared with a variety of other tree species and growth locations around the world⁴⁰. We thus consider it more likely that the ambiguous result is caused by the narrowness of the tree-ring widths (<0.5 mm) in this time which hampered a fully distinct dissection of adjacent rings.”

See also the response to one of the minor comments below, where we write:

“As there was some ambiguity in the detection of the 5259 BCE event in the calendar dated portion of the Bristlecone Pine chronology, which extends from the present back to 6827 BCE, further work must now be done to rule out any possible dendrochronological dating error specific to the 5259 BCE period or to confirm the hypothesis that the thinness of the rings introduced ambiguity to the result. Once this is done, additional single year radiocarbon and dendrochronological measurements can be combined to complete the annually resolved record from the Methuselah Walk bristlecone pine chronology for 10,399 years.”

Minor comments:

Line 76-78: Here the authors cite the paper by Miyake et al. (2017) and regard the 5480 BC event as "currently considered candidate events", but not listed in Table 1 or discussed in the manuscript. I suggest to briefly note the condition of this event here and why it was excluded from the analyses. Also, I suggest to cite Table 1 here to make it clear which events the authors regard as "candidates".

We thank the Reviewer for pointing that out. The reference to Miyake et al. (2017) and the event presented therein was placed by mistake. The abnormal solar modulation discussed in that paper was indeed not in connection to a SEP event. We confused this with the Miyake et al. (2021) paper, which is already cited (reference 22). We have now removed the erroneous citation and added the reference to Table 1.

Line 82-84: The continuous record may only be available for the past 2030 years; however, there are a few floating records as are shaded in Figure 4 or some other records that are sensitive enough to detect past SEPs but not included in IntCal20. Please give a brief but comprehensive overview about the availability of the high-precision data.

This is likely caused by confusion, we apologize for not being clear enough. The 2030 years of annual measurements indicated in Figure 4 actually include all those floating parts. To the best of our knowledge, all published annually resolved measurements are also included in IntCal20, which we used as a basis for our statistical analyses.

To avoid confusion we now write:

“So far, only a few sequences with a total of 2030 yrs of the past 12,400 years (16.5%) have been analyzed with ^{14}C at the annual or biennial resolution, which is required to unambiguously detect SEP events, revealing five strong SEP events (Figure 4).”

Line 89-90: It is not clear, especially to the readers who are not familiar with this field, if the 5-10 yr resolution record is the same as the above mentioned 12,460-yr long record. Include the information on the span of the record if the authors meant any specific record.

We have solved this issue by stating the resolution of the different records individually.

We added more information in this paragraph by writing:

“The motivation to investigate the annually resolved ^{14}C signal in tree-rings between 7150 BCE and 7200 BCE came from lower resolution, synchronized^{29,30} ^{10}Be and ^{36}Cl data in ice cores³¹⁻³³ with a resolution of 5 to 20 years and from decadal ^{14}C data (IntCal20)²⁷.”

Line 159: It was not clear based on what the specific number "beyond 6827 BCE" came from.

We intended to say that the absolutely dated portion of the bristlecone pine chronology reaches back to 6827 BCE. We revised the whole section to improve the clarity of our argument:

“The results of this analysis clearly showed the 7176 BCE event, placing this previously floating sequence on an absolute timescale for the first time. As there was some ambiguity in the detection of the 5259 BCE event in the calendar dated portion of the Bristlecone Pine chronology, which extends from the present back to 6827 BCE, further work must now be done to rule out any possible dendrochronological dating error specific to the 5259 BCE period, or to confirm the hypothesis that the thinness of the rings introduced ambiguity to the result. Once this is done, additional single year radiocarbon and dendrochronological measurements can be combined to complete the annually resolved record from the Methuselah Walk bristlecone pine chronology for 10,399 years.”

Other comments:

1. The manuscript is overall well written; however, I'm concerned about the overuse of parentheses. There are even places where parentheses are unnecessary. Avoid using parentheses and write down in concrete terms. There are places it is not easy to understand.

We now avoid parentheses whenever possible.

2. Figure 4: Indicate the year of the events, otherwise it is difficult to identify which of the events the data points are denoting. Please also increase the number of ticks on the X-axis.

We thank the Reviewer for the suggestion to improve the figure. We now increased the number of ticks on the x-axis and additionally labelled the data points to make clearer to which event they belong.

3. Line 122 and some others: (amount of) excess ^{14}C -> (amount of) ^{14}C excess

We corrected this.

4. Line 134: excess ^{14}C production -> excess of ^{14}C production

We corrected this.

5. Line 277, 287, 291, and 298: EDFig.4a -> Supplementary Fig.4a?

EDFig is changed to Supplementary Fig

Reviewer #2 (Remarks to the Author):

This paper reports observations of ^{14}C increases indicating major solar energetic particle events at two historical periods. The paper is well written and comprehensive. The methods and analysis are appropriate, well executed, and clearly described. It is an important contribution to the field. I find no significant problems with the manuscript and I believe it should be published with only a few minor edits, noted below.

We thank the Reviewer for this positive and supportive review. We were able to address all the minor concerns.

On lines 277, 287, and 291 there are figures referred to by "EDFig"; these refer to figures in the supplementary material and I do not know what is meant by "ED".

EDFig was changed to Supplementary Fig

Reference 8 appears to be missing information and should be completed.

We have updated the reference such that the full title is shown now.

Reviewers' Comments:

Reviewer #1:

Remarks to the Author:

I am still concerned about the discrepancy in the records by the Bristlecone Pine and the Oak/Larch regarding the 5259 BCE event. In the replies and the revised manuscript, the authors raise the consistency between the bristlecone pine and oaks/larch on the 774/775 CE and 993/994 CE events or on the 7176 BCE event to rule out the possibilities that the discrepancy has come from the difference in the tree species. It is however misleading. For example, in the cases SEPs occurred in summer, as was certainly the case for 774/775 CE (see Büntgen et al., 2018; Uusitalo et al., 2018), the signals can be detected in bristlecone pine and in oak/larch in the same year. However, it is possible that the signals may delay in oak/larch in the case the events were in winter or spring (the seasons raised here is just a possible example and might be narrower). For example, the signal of 660 BCE in oak has shown some delay (see Sakurai et al., 2020). I am aware that the separation problem is also possible, but all of the possibilities should be adequately addressed. Agreement between the two species on the other events than 5259 BCE do not rule out this possibility. Line 119-126 should therefore be modified.

Detailed reply to the reviewer's comments

The reviewer's comments are displayed in black while our replies are written in blue.

Reviewer #1 (Remarks to the Author):

I am still concerned about the discrepancy in the records by the Bristlecone Pine and the Oak/Larch regarding the 5259 BCE event. In the replies and the revised manuscript, the authors raise the consistency between the bristlecone pine and oaks/larch on the 774/775 CE and 993/994 CE events or on the 7176 BCE event to rule out the possibilities that the discrepancy has come from the difference in the tree species. It is however misleading. For example, in the cases SEPs occurred in summer, as was certainly the case for 774/775 CE (see Büntgen et al., 2018; Uusitalo et al., 2018), the signals can be detected in bristlecone pine and in oak/larch in the same year. However, it is possible that the signals may delay in oak/larch in the case the events were in winter or spring (the seasons raised here is just a possible example and might be narrower). For example, the signal of 660 BCE in oak has shown some delay (see Sakurai et al., 2020). I am aware that the separation problem is also possible, but all of the possibilities should be adequately addressed. Agreement between the two species on the other events than 5259 BCE do not rule out this possibility. Line 119-126 should therefore be modified.

The reviewer, as we understand, is implying that the photosynthesis process in indecudate (evergreen) trees can occur year-round and therefore C-14 spike in tree rings may appear earlier in tree tissues if a sharp increase in C-14 in the atmosphere occurred at a time when deciduous trees were not photosynthesizing due to lack of leaves. Photosynthesis can go on in winter when conditions are favorable as well. However, formation of annual rings in spite of this in a seasonal climate is discrete even in evergreen trees, i.e. only in the growing season. Therefore, if we assume that the spike in radiocarbon in the atmosphere occurred in the winter of 5260-5259, it could in no way affect the content of C-14 in the cellulose of ring formed in 5260, but only the next tree ring, just like in deciduous species.

If the period of tree-ring formation in the pine ended later than in the oak and larch then it is possible that a sharp increase in C-14 in the atmosphere in the late summer or autumn of 5260, after the end of the ring formation period in Ireland, the Alps and the Russian north, but before the end of tree ring formation in California may not have affected larch and oak from more northern/highland areas, but would have resulted in an increased C-14 content in the 5260 pine ring. But if this is the case, it is only a consequence of the longer or later onset / cessation of tree ring formation.

Phenological studies of bristlecone pine in a number of growth locations (Hallman and Arnott 2015, Ziaco and Biondi 2016) highlight the variability of growth season in bristlecone pine with age, site specific variables and localized weather events. The most applicable of these (Hallman and Arnott 2015) to the sample used in this study, indicate a growth period of ca. 45-60 days starting once spring temperatures are sufficiently warm, a process that can be delayed when spring temperatures are cool and snow cover remains on the ground for longer than usual. The generally agreed growth season is June/July/August based on these very limited studies, however more recent research (not yet published) does seem to indicate possibilities for growth into September and even October in more recently formed rings. For comparison, the formation of tree rings in larch from Russia (the Yamal Peninsula) nowadays ends at the end of August - beginning of September.

However, we do not think that the difference in timing would have a significant effect. One should take into account that "the event intensities from various locations show a strong correlation with the latitude, demonstrating a particle-induced ^{14}C poleward increase" (Uusitalo, Arppe et al. 2018).

So the supposedly longer duration of the ring formation in California can be compensated for by its much more southern location (30 degrees further south than the larch on Yamal). For example, the 774 event, which is assumed to have occurred in July 774 (Büntgen et al., 2018), i.e. even earlier than in autumn, did not lead to an increase in C-14 in pine in California. But did lead to some increase in C-14 in larch rings formed in 774 in Yamal (see figure compiled from Büntgen et al., 2018 data):

Thus, in our opinion, a more likely explanation for the shift in data for pine in the USA compared to data for other species from other areas is the difficulty of dissecting extremely narrow rings. The assumption that the difference is due to a longer period of ring formation in a pine tree in California is not supported by the 774 data. The fact that the ring-widths in the bristlecone pine are unusually narrow in this period also does not support an extended growth season, although a later onset growth period cannot be ruled out. The reviewer's reference to an event around 660 BC in Sakurai et al, 2020 is not fully conclusive. It is not clear from the results of that paper whether the oak tree is delayed or vice versa.

Nevertheless since there are these uncertainties that we cannot fully resolve at the moment, we now provide a detailed discussion of all the possibilities:

The 5259 BCE event (Figure 1b) was found in four absolutely dated tree-ring chronologies from the Alps, Ireland, Russia and the USA showing an average increase in atmospheric ^{14}C concentrations of $(19.1 \pm 0.6) \%$. We note that the USA Bristlecone record shows a potential early increase in 5260 BCE. This effect could either be because very narrow the tree rings widths (<0.5 mm) around this time hamper full distinct dissection of adjacent rings, or because of a later on-set (and cessation) of growth season in Bristlecone pine, although the narrow tree ring widths by themselves indicate that conditions for growth were difficult during this time. Additionally, with about 50 days, modern growth periods of Bristlecone pines in the White Mountains of California are comparably short (Fritts 1969, Hallman and Arnott 2015). Furthermore, the end of today's growing seasons of Bristlecone pine (end of August, (Hallman and Arnott 2015, Ziaco and Biondi 2016)), Alpine larch (October (Rossi, Deslauriers et al. 2008)), Irish oak (October, (Pilcher 1995)) and Siberian larch (end of August) are not significantly different. However, we still cannot exclude the possibility that in 5260 BCE the peak wood formation for the bristlecone was slightly later in the year than for the other species. If the ^{14}C event occurred in late summer or autumn of 5260 BCE, i.e. after the end of the ring formation period in Ireland, the Alps and the Russian north, but before the end of tree ring formation in California, this

would result in an increased ^{14}C content in the 5260 BCE pine tree ring but not in the other chronologies. If this scenario can be confirmed by future replicate measurements of Bristlecone pine the earlier increase would confine the timing of the event to the end of summer/autumn 5260 BCE. However, a strong correlation of ^{14}C event intensities with latitude has been observed previously (Uusitalo, Arppe et al. 2018). We also observe the smallest ^{14}C increase at lowest latitude in the Bristlecone pine for both events. As a consequence, the ^{14}C amplitude recorded in the 5260 BCE ring of Bristlecone pine would at least partly be compensated by its much more southern location (30 degrees further south than the larch on Yamal). Also, the fact that the event occurred during the Holocene climate optimum which was characterized by a weaker latitudinal temperature gradient (Routson, McKay et al. 2019) that would probably have led to more synchronous growing seasons makes the above scenario less likely. Nevertheless, presently we cannot rule out the possibility that the early increase in Bristlecone pine was caused by the narrowness of the tree-ring widths (<0.5 mm), which might have hampered a fully distinct dissection of adjacent rings, and we consider this the most likely explanation considering the discussion above.

Hallman, C. and H. Arnott (2015). "Morphological and Physiological Phenology of *Pinus longaeva* in the White Mountains of California." Tree-Ring Research **71**(1): 1-12.

Pilcher, J. R. (1995). "Biological considerations in the interpretation of stable isotope ratios in oak tree-rings." Paläoklimaforschung **15**: 157-161.

Rossi, S., A. Deslauriers, T. Anfodillo and M. Carrer (2008). "Age-dependent xylogenesis in timberline conifers." New Phytologist **177**(1): 199-208.

Routson, C. C., N. P. McKay, D. S. Kaufman, M. P. Erb, H. Goosse, B. N. Shuman, J. R. Rodysill and T. Ault (2019). "Mid-latitude net precipitation decreased with Arctic warming during the Holocene." Nature **568**(7750): 83-87.

Uusitalo, J., L. Arppe, T. Hackman, S. Helama, G. Kovaltsov, K. Mielikäinen, H. Mäkinen, P. Nöjd, V. Palonen, I. Usoskin and M. Oinonen (2018). "Solar superstorm of AD 774 recorded subannually by Arctic tree rings." Nature Communications **9**(1): 3495.

Ziaco, E. and F. Biondi (2016). "Tree growth, cambial phenology, and wood anatomy of limber pine at a Great Basin (USA) mountain observatory." Trees **30**(5): 1507-1521.

Reviewers' Comments:

Reviewer #1:

Remarks to the Author:

I apologize that the previous comment on the possible impact of tree species on the differences in the carbon-14 profiles lacked detailed information. Although there may be some impact from the growth seasons of the trees as added by the authors to the manuscript, the more important factor is the difference in the time lag between the photosynthesis and the usage of the photosynthate between the deciduous and indeciduate trees. In the case of deciduous trees, due to the lack of leaves at the early phase of growth season, they use the photosynthate produced in the previous year. It is also reported that the photosynthetic activity of some deciduous trees may peak in autumn (Burnett et al., 2021), implying that the majority of the photosynthate used for the growth could be those produced in the previous year. These two factors can cause the delay of the rapid signal in deciduous tree by 1 year. For example, in the case the solar event occurred in August when the growth of the trees is more or less finished but the photosynthesis of deciduous tree is active (or, to be most active onward), signals appear next year both in deciduous and indeciduate trees. However, if the event was in e.g. spring, it is possible that the signals appear in indeciduate tree without delay but they are delayed by 1 year in deciduous trees.

I do not deny the possibility that the difference has come from the separation problem, but the above feature can also explain the difference. I hope the authors include this aspect in the manuscript.

Reference:

Burnett et al., Seasonal trends in photosynthesis and leaf traits in scarlet oak, *Tree Physiology*, 2021.

Detailed reply to the reviewer's comments

The reviewer's comments are displayed in black while our replies are written in blue.

Reviewer #1 (Remarks to the Author):

I apologize that the previous comment on the possible impact of tree species on the differences in the carbon-14 profiles lacked detailed information. Although there may be some impact from the growth seasons of the trees as added by the authors to the manuscript, the more important factor is the difference in the time lag between the photosynthesis and the usage of the photosynthate between the deciduous and indecudate trees. In the case of deciduous trees, due to the lack of leaves at the early phase of growth season, they use the photosynthate produced in the previous year. It is also reported that the photosynthetic activity of some deciduous trees may peak in autumn (Burnett et al., 2021), implying that the majority of the photosynthate used for the growth could be those produced in the previous year. These two factors can cause the delay of the rapid signal in deciduous tree by 1 year. For example, in the case the solar event occurred in August when the growth of the trees is more or less finished but the photosynthesis of deciduous tree is active (or, to be most active onward), signals appear next year both in deciduous and indecudate trees. However, if the event was in e.g. spring, it is possible that the signals appear in indecudate tree without delay but they are delayed by 1 year in deciduous trees.

I do not deny the possibility that the difference has come from the separation problem, but the above feature can also explain the difference. I hope the authors include this aspect in the manuscript.

Reference:

Burnett et al., Seasonal trends in photosynthesis and leaf traits in scarlet oak, *Tree Physiology*, 2021.

We apologize for not having included a discussion of the physiological differences between deciduous and indecudate trees. We added a discussion on a possible delay of the signal in deciduous trees due to different usage of the photosynthate. The reason why we did not include the proposed explanation by reviewer #1 is (and we noted this before), that we didn't see the proposed behavior this on previously analyzed events. Specifically, the Bristlecone pine measured for the 774/775 AD event (that most likely happened in June – August) did not pick up the signal already in 774 AD, while some other (deciduous) trees did (see figure below). Reviewer #1 proposes the opposite. Having said that, we agree with the reviewer that we should be open to any possible examination and we must also consider we cannot necessarily conclude from one year to another. Nature is not that simple. We adapted the text also including the proposed explanation by the reviewer. We also moved this whole discussion (it was originally not considered a discussion, but now it clearly is) into the discussion section of the manuscript, where we think it is better placed.

We now write:

“For the 5259 BCE event, we note that the bristlecone record shows a potential early increase in 5260 BCE. There are a number of possible causes for this. Very narrow ring widths (<0.5 mm) during this period, indicating difficult environmental conditions, may have hampered the complete dissection of the rings, or there may be a previously undetected dating error in this earlier calendar dated portion of the record, where cross-checking with other site chronologies is not possible. Alternatively, this anomalous result may relate to regional shifts in growing season or physiological

differences between deciduous and indecudate trees. If, for example, the ^{14}C event occurred in late summer or autumn of 5260 BCE, i.e. after the end of the ring formation period in Ireland, the Alps and the Russian north, but before the end of tree ring formation in California, this would result in increased ^{14}C content of the 5260 BCE bristlecone pine tree ring, but not in the other chronologies. However, this cannot be argued on the basis of published information on the end of today's growing seasons (Bristlecone pine - late August^{43,44}, Alpine larch - October⁴⁵, Irish oak – October⁴⁶, Siberian larch-late August). Furthermore, the fact that the event occurred during the Holocene climatic optimum, which was characterized by a weaker latitudinal temperature gradient⁴⁷, may mean that more synchronous growing seasons would be more likely. Another possibility may relate to the fact that Bristlecone Pine is the only indecudate tree in this record. Deciduous trees such as oak and larch store photosynthates produced during the end of growing season to grow the next year's earlywood⁴⁸. If the ^{14}C event occurred towards the very end of the deciduous tree's growth season it is possible that only the indecudate Bristlecone Pine would register the change in the same year. If this early increase of ^{14}C in Bristlecone Pine can be confirmed by future replicate measurements, the timing of the event would be confined precisely to the end of summer/autumn 5260 BCE."

Kagawa, A., A. Sugimoto and T. C. Maximov (2006). " $^{13}\text{CO}_2$ pulse-labelling of photoassimilates reveals carbon allocation within and between tree rings." Plant, Cell & Environment **29**(8): 1571-1584.

Routson, C. C., N. P. McKay, D. S. Kaufman, M. P. Erb, H. Goosse, B. N. Shuman, J. R. Rodysill and T. Ault (2019). "Mid-latitude net precipitation decreased with Arctic warming during the Holocene." Nature **568**(7750): 83-87.

Uusitalo, J., L. Arppe, T. Hackman, S. Helama, G. Kovaltsov, K. Mielikäinen, H. Mäkinen, P. Nöjd, V. Palonen, I. Usoskin and M. Oinonen (2018). "Solar superstorm of AD 774 recorded subannually by Arctic tree rings." Nature Communications **9**(1): 3495.

Figure: Shown in blue are the values measured for the Bristlecone pine tree compared to the average. The value is not yet increased in 774 AD (when the event most likely happened) in contrast to average of all 26 trees measured for the Northern Hemisphere in the Buntgen et al. 2018.

Buntgen, U. *et al.* Tree rings reveal globally coherent signature of cosmogenic radiocarbon events in 774 and 993 CE. *Nat Commun* **9**, 3605, doi:10.1038/s41467-018-06036-0 (2018).